# Grip force control under sudden change of friction

Laurence Willemet[1,2] , Felix Roël[1], David Abbink[1], Ingvars Birznieks[3,4] and Michaël Wiertlewski[1]

[1]*Delft University of Technology, Delft, The Netherlands*
[2]*Massachusetts Institute Technology, Cambridge, Massachusetts, USA*
[3]*NeuRA, Randwick, New South Wales, Australia*
[4]*School of Medical Sciences, University of New South Wales, Sydney, New South Wales, Australia*

Handling Editors: Richard Carson & Vaughan Macefield

The peer review history is available in the Supporting Information section of this article (https://doi.org/10.1113/JP286486).

**The Journal of Physiology**

**Abstract figure legend** This study explores how humans adapt their grip force to sudden changes in friction and/or load during object manipulation. Using a novel friction-modulating device integrated into an object suspended by a pulley system, we measured participants' grip force across three phases: lifting, holding, and reacting to perturbations. The results reveal how the sensorimotor system finely tunes grip force in response to grip safety requirements presented on the table on the right.

**Laurence Willemet** received her master's degree in 2017 in technology applied to health in Strasbourg in France, where she worked on reducing surgeons' fatigue during a robot-assisted suture task. She obtained her PhD at Aix-Marseille Université in France, focusing on the role of the sense of touch in manipulation. Her research investigated how the soft skin tissues deform to enable a rapid sensation of friction on initial contact and during incipient slippage in humans. It received the recognition of the Eurohaptic Society PhD Award and has been published by Springer. Following her PhD, she joined the Cognitive Robotics team at Delft University of Technology in the Netherlands for a postdoctoral position, where she developed gentle-touch capabilities for robots to enable delicate tasks such as fruit picking. She is now a postdoctoral fellow within the Computer Science and Artificial Intelligence Laboratory (CSAIL) at MIT. Her current research focuses on bilateral tactile telemanipulation to bridge the gap for touch over a distance.

**Abstract**   A task as simple as holding a cup between your fingers generates complex motor commands to finely regulate the forces applied by muscles. These fine force adjustments ensure the stability and integrity of the object by preventing it from slipping out of grip during manipulation and by reacting to perturbations. To do so, our sensorimotor system constantly monitors tactile and proprioceptive information about the force object exerts on fingertips and the friction of the surfaces to determine the optimal grip force. While the literature describes the transient responses, humans can generate to react to perturbations in load force, it is yet to be determined if humans can also react to abrupt changes in friction while already holding an object. Only recently technology using imperceivable ultrasonic vibrations became available to modulate friction in real time to investigate this question. In this study, we used an object with an integrated friction modulation device suspended in a pulley system controlling the load. With this device, we explored the rapid adaptation of the sensorimotor system to changes in friction alone and in combination with changes in load. When load force and friction changed simultaneously, the grip force response was regulated based on the grip safety requirements. Participants increased their grip force in response to decrease in friction. However, they did not adjust their grip force when the friction increased, which is expected based on our biomechanical model of friction sensing mechanisms.

(Received 4 March 2024; accepted after revision 12 November 2024; first published online 14 December 2024)

**Corresponding author**  L. Willemet: Delft University of Technology, 2628CD Delft, The Netherlands. Email: lwilleme@mit.edu

**Key points**

- Simple tasks like pouring water into a glass mobilize intricate interactions between fingertip sensory inputs and motor commands to account for the weight change and friction.
- It has been investigated how humans react to force perturbations when holding an object, but very little is known about how frictional changes are sensed and acted upon while holding an object, for example, due to sweating or condensation.
- We engineered a unique experimental object that utilizes imperceivable ultrasonic vibrations to change the frictional properties of the surface in a few milliseconds. This apparatus enabled us to study how human subjects react to change of friction when gripping or holding an object.
- We showed that humans adjust the strength of their grasp when forces in the direction of gravity either increase or decrease; however, frictional change evokes adjustments only when friction decreases.

# Introduction

Peeling a juicy peach without squashing it and preventing it from slipping and falling out of hands can be a challenge especially when juice is making the surface more slippery. The friction between fingertip skin and the object during manipulation can vary due to several factors such as sweating and condensation. Our nervous system must adapt to this highly variable frictional interaction and react to sudden changes in friction. Despite dynamic changes in various parameters to be controlled during dexterous manipulation, humans display a remarkable ability to precisely adjust the force exerted on the object, taking into account the weight and surface properties of the object, as well as the nature of the task. The control of

grasp stability encompasses the prevention of accidental slips and the prevention of excessive forces. A key aspect of this control involves adjusting grip forces to the friction such that a destabilizing load force, pulling the object out of the grip, is counteracted by the frictional force. The fine control allows humans to hold objects securely, balancing the forces to avoid slip and at the same time to avoid damage to a fragile or deformable object.

To adjust the grasp, the motor system uses multiple control strategies, relying upon available sensory input (Khamis et al., 2014b), predictive anticipatory mechanisms (Flanagan & Beltzner, 2000), optimal actions strategies (Hadjiosif & Smith, 2015), error correction (Johansson & Westling, 1988) combined with information provided by indirect associative sensory

cues (Buckingham et al., 2009; Witney et al., 2004). There is ample evidence in the literature supporting that grip forces are adjusted to the surface properties, especially the amount of frictional strength of the surface/finger contact. These adjustments are observed after about 100 ms of initial contact with an object, even in the absence of lateral force on the fingertip (Johansson & Westling, 1984; Westling & Johansson, 1984). After initial contact with the surface, the object is lifted upwards during which the grip force increases in proportion to the load force and the slipperiness of the object (friction). During holding an object, the nervous system is constantly regulating grip force to avoid slippage. It is proposed that it does so by detecting partial slips and monitoring the fraction between the contact area in safe contact with the skin and the area subjected to slippage, denoted as the 'stick ratio' (Delhaye et al., 2024; Khamis et al., 2014b; Tada, 2006).

The grip force adjustments are likely triggered by early signs of incipient slippage of the object in contact with the skin (Afzal et al., 2024, 2022). At a mechanical level, when grasping an object, the skin experiences a compression on the trailing edge and a dilatation on the leading edge of the contact area (Delhaye et al., 2016; Schiltz et al., 2021). It has been previously shown that those deformations are directly dictated by the weight of the object as well as the amount of friction available at the interface (Willemet et al., 2022).

In early studies on grip regulation, it was unclear whether humans adapted their grip force by sensing friction or using texture as a cue associated with expected frictional change (Johansson & Westling, 1987). Other experiments explored whether texture or friction was responsible for the adjustment (Cadoret & Smith, 1996). In this study, texture cues were not available, but due to use of lubricants friction was changed after a block of 10 trials, and thus was predictable and the grip force adjustment could have been achieved by anticipatory mechanisms.

Recently ultrasonic programmable friction modulation devices offer a way to overcome these experimental limitations. This technology uses imperceivable vibrations at ultrasonic frequencies to modify the contact of the fingertip skin and instantaneously change friction of a glass surface (Biet et al., 2007; Winfield et al., 2007; Wiertlewski et al., 2016). This new instrument offers a programmable control over the frictional strength without providing other cues as surface material remains the same and there is no need to apply any lubricants or grip enhancing compounds. Only recently its full potential was revealed when investigating friction perception mechanisms in the context of object manipulation (Afzal et al., 2022, 2024; Khamis et al., 2021). We observed that during initial contact with an object, the frictional information can become available before any lateral movement occurs (Khamis et al., 2014a; Willemet et al.,

2021). It has never been used to observe grip force adjustments and investigate mechanisms in motor tasks. Because friction is a central mechanism in grasping, we hypothesized that the nervous system can react to rapid changes of friction, especially when the object becomes more slippery.

In this study, we leverage this unique technology to investigate how sensory information about friction changes when holding an object is used by the motor system. Specifically, we were able to investigate the effects of increasing or decreasing friction, on its own and in combination with load force perturbations. Another advantage of changing friction while the participant is holding an object is that it allows us to observe adjustments to sudden relative friction change more clearly than if comparison would have been made between trials subjected to unpredictable frictional variations. Frictional adjustments within one trial are physiologically relevant as friction can significantly change between the moment when initial touch occurs (Tomlinson et al., 2009) and later stages of manipulative task (André et al., 2010). From the sensory systems point of view, the adjustments to frictional change while holding an object are more challenging without the contribution of reaching and grasping movement kinematics to friction sensing (Afzal et al., 2024).

## Materials and Methods

### Participants and protocol

Twenty right-handed volunteers (16 men and 4 women, 24.7±2.3 years old) participated in the study. The study was approved by the ethics committee of the TU Delft (HREC 1722) and the study conformed to the standards set by the Declaration of Helsinki, except for the registration in the database. Participants gave their informed written consent before participating in the experiment. The participants were naive to the hypotheses posed in this study, and they did not report having any skin conditions, perceptual or motor deficits which might influence the study outcomes. They were asked to accomplish 36 testing trials after 12 trials of familiarization with the set-up. In each trial, they should casually lift, hold and replace the object using the tips of their index finger and opposing thumb of their right hand using only as much force as necessary. They were instructed to refrain from paying more attention than they typically would when casually gripping everyday objects.

During pilot experiments, we noticed that participants could not help but pay too much attention to the complex looking instrumented object. To be able to observe and study reflexive behavioural reactions, we diverted their attention by asking participants to watch a nature documentary while manipulating the object. The

participants wore headphones playing the audio of the documentary which they watched on the screen to divert their attention from the task and later were quizzed on. To prevent the influence of auditory cues, noise-cancelling headphones were used (3M, HRXS220A).

Each trial was split into the following three phases: a lifting phase, a holding phase and a reacting phase (Fig. 1*A*). During the lifting phase, participants encountered the following two different conditions: a neutral friction and an ultrasonically reduced friction where the amplitudes of the driving signals were maximum. Before continuing to the holding phase, the participant's grip on the object was required to be settled (i.e. both the velocity and the grip force variance should be lower than $5 \cdot 10^{-3}$ m/s and $1 \cdot 10^{-3}$ N/s, respectively) and to be below 5 N to make sure the potential friction perturbation magnitude remains relatively high (Fig. 1*B*). When the object was considered stationary based on the velocity threshold and the grip force exceeded the upper bound, a sound was played to warn participants to reduce their grip force.

To limit predictive grip responses in anticipation of a perturbation, the start of the holding phase was not cued to the participant. After 2 s, the participant's grip on the object was perturbed in load force and/or friction. Nine different perturbation conditions were presented in random order, in which the load force and friction were either decreased, kept constant or increased (Fig. 1*C*). Ideally, the grip force would be adjusted as a function of both the load force and the friction changes. Both the load force and the friction modulation signal amplitude changes were ramped in 0.05 s. During the experiment, forces were acquired from a force sensor (ATI, Nano 43) and vertical position of the object from an incremental encoder (Baumer, BTIV 24S 16.24K 1024 G4 5) (see Fig. 2*B*).

## Experimental set-up

The instrumented object was guided on a 8-mm steel shaft (Bosch Rexroth, 400 mm) using two linear ball bearings (INA, KH08-B) to counteract any torques that arise around the fingertip-object contacts. A fully circular shaft was used as a linear guide to allow rotation of the object in the horizontal plane around the guide (Fig. 2*A*) and ensure that normal force balance would not change between fingers due to tilting. The final object measured $78 \times 67 \times 38$ mm³ and weighted 293 g, which made it easy to hold with two fingers in a pinch grasp.

The apparent weight of the object could be programmed by pulling the object in the direction of the gravitational field using two DC motors (Faulhaber, 2642 012 CR). The DC motors delivered a force to the object by means of a capstan transmission where the cable is spooled on the shaft. The pulling force produced by the DC motors was controlled to −0.5, 0 or 0.5 N with a 10-N/s load rate using a servo amplifier (Maxon, LSC 30/2).

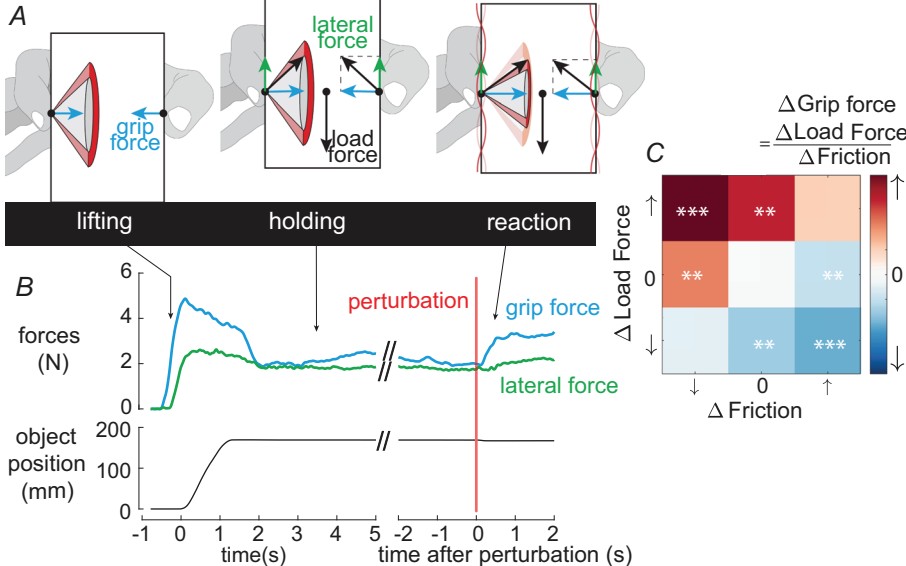

**Figure 1. Typical trial of a grip force response to a friction decrease**

*A* The three main phases during the task: lifting, holding and reacting to change of friction and/or load force. The grey cone shows the admissible forces before sliding that ensure more than 20% safety margin. The red cone is the cone of friction, whose edges correspond to the forces that will lead to object slippage. *B* Lateral force measured for one plate, grip force and position of the object are shown for a typical trial. Curves are synchronized on the lift onset. *C* Ideal grip force adjustments expected upon changes in load force or friction. * indicates significance. [Colour figure can be viewed at wileyonlinelibrary.com]

The object was equipped with two friction modulation devices that could change the frictional condition within 20 ms. The friction modulation device consists of a glass plate, which vibrates at ultrasonic frequencies with a flexural standing wave. The vibration of the plate induces a micrometric levitation of the skin of the fingertip, thereby reducing the interfacial friction. The rectangular glass plate ($68 \times 52 \times 5$ mm$^3$) vibrated at a frequency of 29.97 and 30.06 kHz (respectively for the index and the thumb plate) with a mode shape with three nodal lines. Each friction modulation plate was actuated by an array of two piezoelectric actuators (Steminc, SMPL-26W16T07111), glued on the non-contacted side of the surface (Fig. 2*B*). The signals driving the actuators were chosen to maximize the friction reduction, while preventing heating at the interfaces between friction modulation plates and fingertips. The voltage amplitude of the signal was set to 25 V for the plate that was gripped by the index finger and to 20 V for the plate that was gripped by the thumb, resulting in a maximal settled vibration amplitude of $4.6 \pm 0.17$ $\mu$m for the index finger plate and $4.5 \pm 0.15$ $\mu$m for the thumb plate. These vibrations are absolutely imperceptible as no cutaneous receptors in humans are able to respond to such high frequency and low vibrations amplitude.

The relationship between vibrations amplitude and friction at both plates was measured in one participant who performed 30 sliding movements on the fixed plates (Fig. 2*D*). Friction was changed unpredictably, and no texture cues were available to this participant.

Before the lifting phase, the vibrations amplitude was set to either 0 or 4.5 $\mu$m. Then, the friction was either increased, decreased or kept constant by decreasing the vibrations amplitude to 0 $\mu$m, increasing to 4.5 $\mu$m, or keeping it constant, respectively.

## Data processing and statistical analyses

The fingertip forces were low-pass filtered with a second-order Butterworth filter with a cut-off frequency of 2.7 Hz and synchronized based on the lift onset.

*Initial grip force adjustments to friction before the lift-off.* To evaluate how early during the trial the grip force began to significantly differ between two frictional conditions, a two-sample sequential *t* test (Hajnal, 1961) was performed comparing the grip force applied by participants during the lifting for both frictional conditions (neutral and reduced). When the *t* test decision accepted the H1 hypothesis, we performed successive *t* tests on grip force level measured at each 1.6-ms time bin before the object lift-off. The grip force was considered significantly different between both friction conditions if the *P*-value was lower than 0.05.

*Grip force adjustments after perturbation.* To quantify the grip response to combined changes in friction and load force, we computed the average relative grip force changes as follows:

$$\Delta F_n(t) = \frac{F_n(t)}{\frac{1}{t_p} \int_0^{t_p} F_n(\tau)\, d\tau} - 1 \qquad (1)$$

where $t_p$ is the time at which the perturbation is applied. For each trial, we computed the average magnitude of the relative grip force adjustment as the mean of the grip force that is applied between 100 ms after the perturbation and the end of the holding phase which was 10-s long. This 100-ms latency gives a provision for potential grip reflexes to be triggered (Cole & Abbs, 1988; Johansson & Westling, 1987). The relative grip force response was then normalized by dividing it by the baseline grip force before the perturbation. To investigate the influence of the perturbation on the motor command, we performed

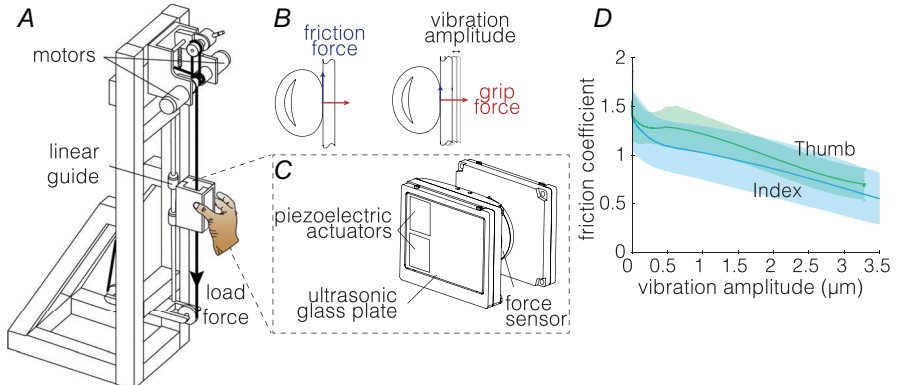

**Figure 2. Schematic and working principle of the experimental set-up**
*A* Schematic of the experimental set-up. The object equipped with friction-reduction glass plates on both sides was free to move along and around the linear guide. *B* When ultrasonic vibrations are applied to the glass plate, the friction is reduced, requiring a higher grip force to hold the object. *C* Close up of fingers touching surfaces of the object manipulated by the participants. *D* Normalized friction coefficients for the thumb and the index finger plates as a function of the amplitude of vibration computed from a series of 30 sliding movements for one subject. [Colour figure can be viewed at wileyonlinelibrary.com]

a statistical analysis (*t* test) in which the normalized grip force after the perturbation was tested in comparison to the normalized grip force during the reference condition, where no perturbation was applied.

Data distribution normality was tested with the Anderson–Darling test. If the null hypothesis was rejected, non-parametric tests were used: the *t* test was replaced with a Wilcoxon signed-rank test and the ANOVA was replaced by a Kruskal–Wallis *H* test.

## Results

In this section, we first report grip force adaptations to friction when lifting an object. Subsequently, we explore how grip force changes in response to external load force perturbations and sudden changes in friction when the object is held in the air.

### Adaptation to friction during lift

First, we assess how subjects adjust their grip forces in response to various friction levels when grasping and then lifting the object. Figure 3*A* shows the grip force for one participant for the neutral and reduced friction condition. We found that the surface friction significantly influenced the grip force exerted by 12 subjects out of 20 ($P < 0.05$ two-sample sequential *t* test). When looking at how they lifted the object under low and high friction conditions, we found that one of them adjusted its grip force after the lift-off and was considered as an outlier; the remaining eleven participants are shown in Fig. 3*B*. The grip force adjustments to friction could be observed before the lift-off after a median time of 363 ms (94–913-ms quartiles) measured from the initiation of contact. The grip force measured at the time point where

high and low friction has significantly different values was 0.18 N (0.10–0.58-N quartiles). The remaining eight subjects employed significantly higher grip forces at the time of the object lift-off ($t(89) = 2.2$, $P = 0.03$). At the object lift-off, the grip forces measured were $3.4 \pm 0.85$ N, compared to $2.6 \pm 0.57$ N for the subjects from the responder group. These grip forces would be sufficient to maintain a safe grip regardless of how slippery the object was.

We wanted to illustrate at what grip force level cutaneous receptors are capable of obtaining frictional information. Assuming that it would take more than 50 ms for the sensory input carrying frictional information to trigger detectable changes in force, we measured the grip force 50 ms before the friction effects could be discerned for each subject. Accordingly, the grip force at which sensory information was estimated to be obtained was 0.13 N (0.071–0.29 quartiles; $N = 11$) or less.

### Response to a perturbation

**Load force perturbation.** To ensure that the object does not slip out of the grasp, any load force applied to the object has to be counteracted by frictional forces. An increase in maximum achievable frictional force can be obtained by increasing the magnitude of the grip force. Grip force response to load perturbations is plotted in Fig. 4*A*, and magnitudes of relative grip force response are shown in Fig. 4*B*. Conditions with significant differences are indicated by asterisks. In the control condition without load force perturbation, participants did not significantly change their grip force while holding the object ($t(119) = -0.964$, $P = 0.34$). For trials with the load force perturbation, we found highly significant differences between the baseline grip

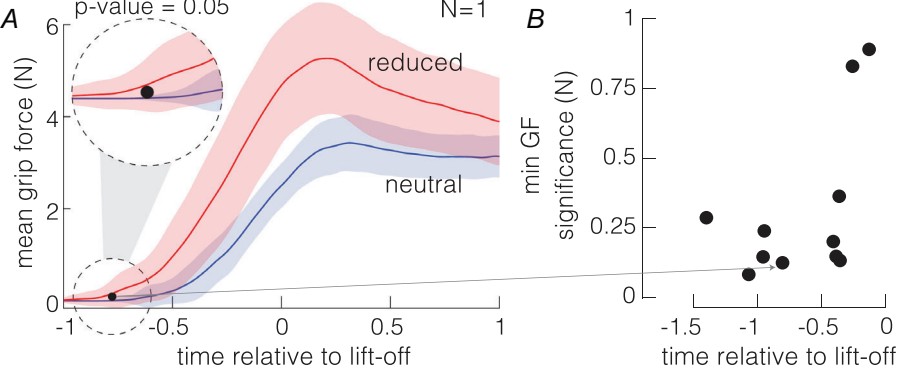

**Figure 3. Grip force adaptation during the object lift**
*A* Average grip force for one participant per friction condition for 18 trials. The grip force was considered to be adjusted to friction when the *P*-value of the two-sample sequential *t* test was lower than the significance level. Red denotes reduced friction, blue – neutral (unaltered) frictional condition. The shaded area denotes SD. *B* Grip force for the first point of significance as a function of the time relative to the lift-off. Each point stands for one participant, *N* = 11 participants show significant difference of grip force between both friction conditions before the lift-off. [Colour figure can be viewed at wileyonlinelibrary.com]

force and grip force level after the load force was increased ($t(119) = 8.01$, $P = 1.8 \cdot 10^{-7}$) or decreased ($t(119) = -4.03$, $P = 2.7 \cdot 10^{-5}$).

If the object is held safely within the grip, then the change in load force may cause the displacement of the object and the hand together to move in the direction of the load force. In our experiments, the load force perturbation had a significant effect on the vertical position of the object $z$ after the load force decreased ($t(119) = 3.90$, $P = 1.6 \cdot 10^{-4}$) or increased ($t(119) = -5.08$, $P = 1.4 \cdot 10^{-6}$). The vertical position was correlated with the magnitude of the grip force response (Pearson correlation: $r(358) = -0.466$, $P = 4.9 \cdot 10^{-19}$) (see Fig. 4C). This result suggests that the larger the displacement of the object, the stronger the participants are reacting in terms of grip force magnitude. The larger displacement magnitude of the object might be associated with an amount of skin stretch, the extent of full slippage or joint compliance of each subject. After the object vertical position changed due to the load force perturbation, no corrective response aimed to minimize deviation was observed. Hence, the vertical displacement is most likely due to the object movement within the grip rather than the movement of the hand or a whole arm at the extent which would evoke corrective responses such as myotatic reflex driven by proprioceptive input.

**Friction perturbation.** In our experiments, we abruptly changed the level of friction while the participants were holding the object. The increase in friction did not induce any grip force response within the 8 s after the perturbation ($t(59) = 1.36$, $P = 0.18$).

However, when friction was decreased, participants applied a significantly higher grip force ($t(59) = 3.96$, $P = 1.7 \cdot 10^{-4}$) to maintain grip stability. There was no indication that this response would be triggered by the object suddenly slipping away and subjects catching it as there were no statistical differences in object vertical position after the friction increased or decreased (Wilcoxon signed-rank, $W = 927$, $z = 0.088$, $P = 0.93$ and $W = 883$, $z = -0.24$, $P = 0.81$) (see Fig. 5A). Thus, the grip force response was initiated by the localized partial slips within the area of contact for objects remaining stationary in the grip.

**Simultaneous friction and load force perturbations.** When friction was decreased and load force increased simultaneously, participants were gripping the object by $16 \pm 21\%$ harder ($t(59) = 5.92$, $P = 1.2 \cdot 10^{-6}$). As expected, this condition had the strongest response because increasing the load force and decreasing the frictional strength would both endanger the stability of the grip the most and thus have the strongest combined effect.

Interestingly, when a decrease in friction was accompanied by a decrease in the load force, the grip force applied by the participants did not change significantly ($t(59) = 1.06$, $P = 0.29$). The participants did not significantly change their grip force because the effect of friction and load force are compensating each other (Fig. 6). This stability might be interpreted that grip force was regulated to maintain a constant safety margin against slips. If so, it would be expected that a simultaneous increase in friction and in the load force would also result in no change in grip force due to the same mutual compensatory effect. However, it was not the case, and subjects reacted to the load force increase regardless of friction increase with an increase in grip force ($t(119) = 5.55$, $P = 9.0 \cdot 10^{-6}$)

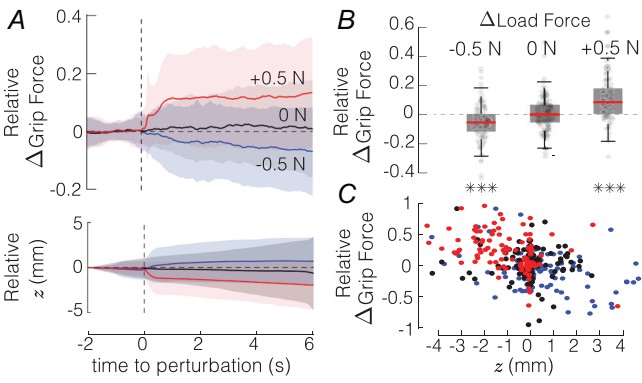

**Figure 4. Response to a load force change**
*A* The relative grip force change and the object vertical position *z* elicited by a load force perturbation. The solid line and shading represent mean ± std (*N* = 20). *B* Magnitude of the relative grip force change for −0.5, and +0.5-N load force change and no change 0 (***P* < 0.001). Boxplots represent median, 25th and 75th quartiles. *C* Relative grip force changes as a function of the vertical position *z* of the object. Blue, black and red dots stand for −0.5, 0 and +0.5-N load force change, respectively. [Colour figure can be viewed at wileyonlinelibrary.com]

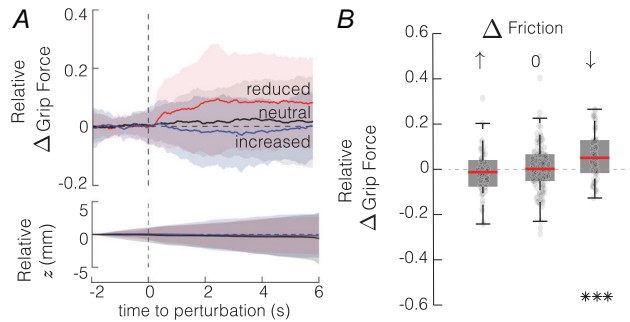

**Figure 5. Response to a friction change**
*A* The relative grip force change and the object vertical position *z* elicited by a friction perturbation (*N* = 20). The solid line and shading represent mean ± std. *B* Magnitude of the relative grip force change with increased, neutral and decreased friction (***P* < 0.001). Boxplots represent median, 25th and 75th quartiles. [Colour figure can be viewed at wileyonlinelibrary.com]

which was as strong as when friction did not change ($t(59) = -1.22$, $P = 0.23$). The discrepancy in these results might be explained by the object movement within the grip – its vertical position did not change when friction and load force were both decreased (Wilcoxon signed-rank $W = 1033$, $z = 0.87$, $P = 0.39$), but it did change when friction and load force both increased ($t(59) = -6.59$, $P = 1.3 \cdot 10^{-8}$). When friction was increased and the load force decreased, then the grip force decreased as expected ($t(59) = -4.24$, $P = 2.5 \cdot 10^{-4}$); however, the magnitude of the grip force response was no different from the equivalent load force decrease when friction did not change ($t(59) = -0.05$, $P = 0.96$). The object vertical position changed by the same amount in both conditions ($t(59) = -0.52$, $P = 0.61$).

**Reaction time.** For the conditions in which we measured a significant grip force adjustment to the perturbation, we computed the maximal grip force rate characterizing response intensity, and the reaction time, defined as the instant when the first derivative of grip force exceeds a defined threshold of 0.8 N/s. We observed the highest grip force rate and the shortest reaction times when the load force increased (Fig. 7). Globally, the load force change had a significant influence on both the maximum grip force rate and the reaction time (Kruskal–Wallis $H$ test, $H(2) = 25.3$, $P = 3.2e^{-6}$ and $H(2) = 11.7$, $P = 0.003$, respectively). Participants reacted within a median time of 192 ms (72-408 ms quartiles) from the time of the perturbation when the load force increased, whereas the load force decrease resulted in the grip force

adjustments (decrease) after about 463 ms on average. When the load force stayed constant, but the friction decreased, the reaction time was slightly longer (274 ms, 189–426-ms quartiles) than for the load force increase but the maximum grip force rate was similar, suggesting that even if participants need a longer time to sense a friction change, they are still responding with the same intensity.

## Discussion

The aim of this study was to demonstrate fingertip grip force adjustments to friction when reaching, gripping and lifting the object. The novelty in this study is that for the first time, we were also able to demonstrate how sudden changes in friction, while holding an object, influence grip force adjustments. Such investigations were enabled by new technologies embedded in the object which can change friction of the gripped surfaces in a controlled way using imperceivable vibrations at ultrasonic frequencies. The advantage over previous studies investigating grip force adjustments when gripping and lifting objects is that friction could be changed unpredictably at any point of time without changing surface material, thus avoiding giving any additional cues which might be associated with frictional change such as texture or presence of lubricant or friction enhancers.

### Adjustments to friction before the object lifts off

Our study demonstrated that when subjects reach and grip an object, the grip force is adjusted to friction before the

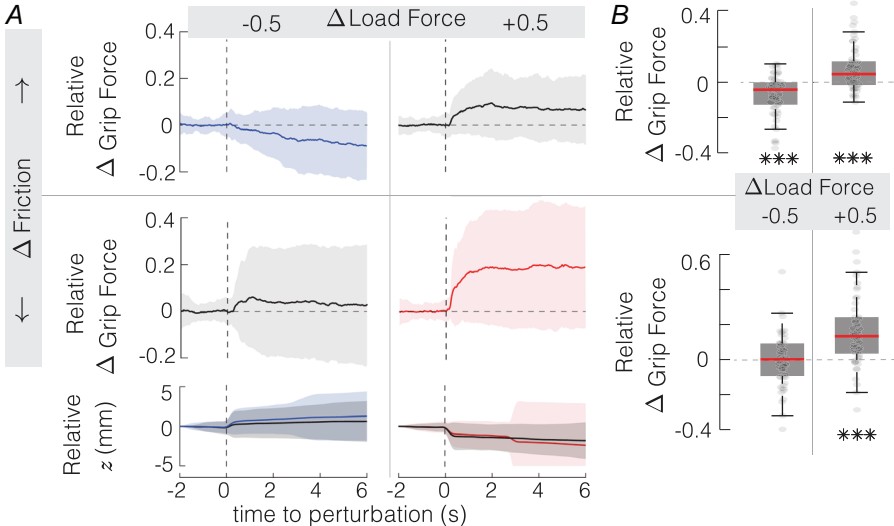

**Figure 6. Response to a similtaneous load force and friction changes**
*A* The relative grip force change elicited by a load force and friction perturbation. Blue and red lines denote conditions in which friction and load force level change both would require a decrease and increase in the minimum required grip force, respectively, to maintain a stable grasp (*N* = 20). The solid line and shading represent mean ± std. *B* Magnitude of the relative grip force change (***P* < 0.001). Boxplots represent median, 25th and 75th quartiles. [Colour figure can be viewed at wileyonlinelibrary.com]

object's lift-off for 11 out of 20 participants (Johansson & Westling, 1984). Thus, despite being asked to apply the lowest force required to lift the object, some participants consistently chose to use grip forces high enough to lift the object regardless of the friction condition. There might be several explanations. Some subjects might have used overall high grip forces being afraid to drop and damage the instrumented object, and it is also possible that differences in strategy and kinematics of reach and grip movement might have influenced the richness of sensory information available at the initial contact (Afzal et al., 2022, 2024).

The importance of the initial contact in friction sensing is supported by an observation that in 11 out of 20 participants who adjusted their grip force to friction before object lift-off, adjustments to friction were detected at very small grip force levels being just 0.18 N on average.

However, the grip force adjustments to friction before the lift-off were not present in all subjects, and the latency was longer than previously reported in studies originating from Johansson's laboratory (Johansson & Westling, 1984) using textured surfaces and on par with other similar recent studies using smooth glass surfaces (Delhaye et al., 2024). Apart from texture, the difference might be also caused by a lower coefficient of friction producing a less pronounced skin stretch, responsible for triggering the response. The time between the first contact and the lift-off (388 ± 248 ms) was also slightly longer, possibly being due to increased care and cautiousness participants showed when manipulating the unusual object equipped with many highly technological components.

### Responses to friction and load force perturbations when holding an object

After the object was lifted and held stationary within a grip, the grip force responses to sudden changes in friction or load force and combination of two were investigated. When only friction was changed, we observed that

a decrease in friction elicited adaptive grip increase response regardless that the object did not move with the grip. This indicates that frictional change was sensed exclusively based on tactile input signalling partial slips within contact area and resulting skin deformation changes. Such grip force increase reaction is likely intended to restore the safety margin to a value that would avoid object slippage. In contrast, an increase in friction did not evoke systematic changes of grip force. Such a result was expected because there are no known biomechanical events at the contact that would suggest how an increase in friction could be signalled by the skin receptors under static conditions. According to a biomechanical model of friction sensing mechanisms at the fingertip (Willemet et al., 2021), the friction is signalled by a skin divergence pattern. Indeed, higher friction will preclude skin from being pushed and radially diverge when pressure differences between central and peripheral portions of the contact area build up as fingertip skin is being compressed during the contact. If friction decreases, the elastic energy is released, and the skin radially diverges – an event which can be signalled by tactile afferents. However, if the skin has already diverged under a low friction condition, now increasing friction would have no effect – it would simply lock the shear strain pattern in the skin. This hypothesis is in agreement with previous experiment reporting that humans are failing to perceive friction increases (Monnoyer et al., 2023). From grip force adjustment standpoint, it is still possible that the motor system could employ a strategy that after a safe grip is established, the grip force strength might be slowly released (Johansson & Westling, 1987) until afferents would start signalling small slips at the periphery of the contact area (Khamis et al., 2014b; Schiltz et al., 2021). This would make sense biologically, because it has been demonstrated that sweating might purposefully increase friction over time and thus would be logical to expect a mechanism taking advantage of it. Nevertheless, participants in this study did not seem to exploit such a strategy, and grip force was kept stable over the time course of 8 s after the friction change.

We would like to emphasize that the full slippage of the object was not required to trigger the responses because we observed that grip forces increased regardless of whether we observed a gross displacement of the object or not. Upon application of a load perturbation, the vertical position of the object changed. However, regardless that the grip force response was triggered, no response which would aim to return the object to its original position was observed. A large part of the object movement might have taken place because of stretching the skin and inducing partial slips within the contact area (Khamis et al., 2014b).

The latency of grip force response to friction decrease was 274 ms (189–426-ms quartiles). In comparison for the increase of load force, participants reacted within

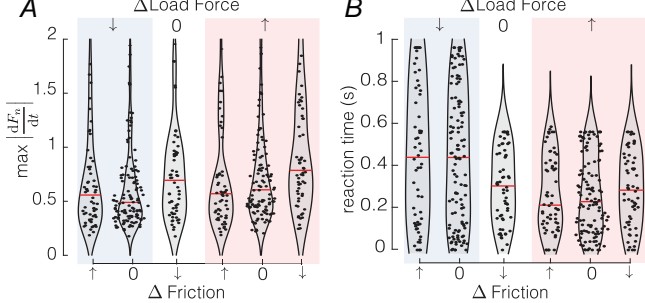

**Figure 7. Reaction times**
*A* Absolute value of the maximum grip force rate elicited by the perturbation (*N* = 20). *B* Reaction time for each perturbation condition (*N* = 20). The red lines indicate median values. [Colour figure can be viewed at wileyonlinelibrary.com]

192 ms (72–408-ms quartiles) after the perturbation. While it is difficult to compare these numbers with the literature because of differences in methodology, we can note that the 100-ms difference measured between the reaction to a load force change and a friction change can signal a difference in processing of the tactile input. It is likely that frictional changes, when already holding an object, primarily activate tactile receptors sensing shear strain changes within the contact area (Khamis et al., 2014a; Macefield et al., 1996). However, the load force perturbation would stretch the skin and activate receptors also outside the contact area between its border and the nail (Birznieks et al., 2009). The load force increase and the friction reduction activate afferents in a different pattern. The load increase creates a gross deformation of the fingertip skin stretching it (sensed by SA-II afferents) and also alters the magnitude of the local strain pattern at the contact area, sensed by SA-I and FA-I afferents. In contrast, when friction is changed, this gross deformation does not occur and therefore the activation of SA-II would be lower. In addition, load force changes might activate proprioceptors (Macefield & Johansson, 1996). It has been demonstrated that when cutaneous input is not available, proprioceptors might contribute to grip reflex, but at longer latencies (Häger-Ross & Johansson, 1996). In our study, we did not observe any evidence pointing to significant contribution of proprioceptive afferents – with friction change, there was no object position change capable of activating proprioceptors. When load force perturbation was applied, no signs of myotatic reflex counteracting object's movement were observed and response latencies were even shorter than for responses triggered exclusively by tactile afferents in friction change condition.

In this study, we wanted to reduce the influence of cognitive processes to study the response in a context close to daily life manipulation, where attention is not directed to the fine-tuning of grasping forces. After finishing their experiment, only a few participants reported perceiving the friction perturbation and, interestingly, they described the sensation as an increase in moisture. The fact that most participants were not aware of friction changes confirmed that adjustments observed in our study do not require cognitive attention. We also conclude that watching the documentary in our study successfully helped to divert attention of participants.

These results show that ultrasonic friction modulation can significantly impact the sensorimotor regulation of grip by influencing the friction coefficient of the skin/glass interface. We believe this technology has a potential to advance research enabling future exploration of neural mechanisms controlling grip force during object manipulation. The refined understanding of the neural mechanisms underlying grip control under frictional change can inspire haptic devices for rendering realistic sensation in virtual reality devices. Provancher and Sylvester (2009) help to design control algorithms for robotic grippers. This knowledge can also provide new approaches to design clinical tests to detect certain functional deficits or pathologies affecting grasp.

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

## Additional information

### Data availability statement

The dataset obtained in this study as well as the code to process the data can be accessed via https://doi.org/10.4121/20217431.v1.

### Competing interests

The authors declare no conflict of interests.

### Author contributions

L.W., F.R. and M.W. designed the work, F.R. performed the experiment, all authors contributed to data analysis and interpretation, L.W. and M.W. drafted the initial manuscript and D.A. and I.B. revised critically for important intellectual content. All authors approved the final version of the manuscript, agree to be accountable for all aspects of the work in ensuring that questions related to the accuracy or integrity of any part of the work are appropriately investigated and resolved and all persons designated as authors qualify for authorship. All those who qualify for authorship are listed.

### Funding

This work was supported by the 4TU Soft Robotics program. L.W. acknowledges the support of the Marie Skłodowska-Curie Actions (Project ReTouch), I.B. acknowledges the support of the Australian Research Council's (ARC) Discovery project Grant

DP230100048 and M.W. acknowledges the Dutch Research Council (NWO) for the Vidi project 19680.

## Keywords

friction, grip force regulation, sensorimotor control, touch

## Supporting information

Additional supporting information can be found online in the Supporting Information section at the end of the HTML view of the article. Supporting information files available:

**Peer Review History**

