## [Peer Review History · The Journal of Physiology]

Grip force control under sudden change of friction

Laurence Willemet, Felix Roël, David Abbink, Ingvars Birznieks, and Michaël Wiertelwski
DOI: 10.1113/JP286486

Corresponding author(s): Laurence Willemet (lwilleme@mit.edu)

Review Timeline:

Submission Date:	04-Mar-2024
Editorial Decision:	09-Apr-2024
Revision Received:	10-Oct-2024
Editorial Decision:	24-Oct-2024
Revision Received:	04-Nov-2024
Accepted:	12-Nov-2024

Senior Editor: Richard Carson

Reviewing Editor: Vaughan Macefield

Transaction Report:

Dear Dr Willemet,

Re: JP-RP-2024-286486 "Grip force control under rapid change of friction" by Laurence Willemet, Felix Roel, David Abbink, Ingvars Birznieks, and Michaël Wiertelwski

Thank you for submitting your manuscript to The Journal of Physiology. It has been assessed by a Reviewing Editor and by 2 expert referees and we are pleased to tell you that it is potentially acceptable for publication following satisfactory major revision.

REVISION CHECKLIST:

We look forward to receiving your revised submission.

Yours sincerely,

Richard Carson
Senior Editor
The Journal of Physiology

REQUIRED ITEMS

- Author photo and profile. First or joint first authors are asked to provide a short biography (no more than 100 words for one author or 150 words in total for joint first authors) and a portrait photograph. These should be uploaded and clearly labelled together in a Word document with the revised version of the manuscript. See Information for Authors for further details.
- You must start the Methods section with a paragraph headed Ethical Approval. If experiments were conducted on humans, confirmation that informed consent was obtained, preferably in writing, that the studies conformed to the standards set by the latest revision of the Declaration of Helsinki and that the procedures were approved by a properly constituted ethics committee, which should be named, must be included in the article file. If the research study was registered (clause 35 of the Declaration of Helsinki), the registration database should be indicated, otherwise the lack of registration should be noted as an exception (e.g. The study conformed to the standards set by the Declaration of Helsinki, except for registration in a database). For further information see: <https://physoc.onlinelibrary.wiley.com/hub/human-experiments>.
- The reference list must be in alphabetical order, rather than numbered, to comply with our Journal format.
- Your manuscript must include a complete Additional Information section, including competing interests; funding; author contributions and acknowledgements.
- Please upload separate high-quality figure files via the submission form.
- Please ensure that the Article File you upload is a Word file.
- Papers must comply with the Statistics Policy: https://jp.msubmit.net/cgi-bin/main.plex?form_type=display_requirements#statistics.

In summary:

- If $n \leq 30$, all data points must be plotted in the figure in a way that reveals their range and distribution. A bar graph with data points overlaid, a box and whisker plot or a violin plot (preferably with data points included) are acceptable formats.
- If $n > 30$, then the entire raw dataset must be made available either as supporting information, or hosted on a not-for-profit repository, e.g. FigShare, with access details provided in the manuscript.

- 'n' clearly defined (e.g. x cells from y slices in z animals) in the Methods. Authors should be mindful of pseudoreplication.
- All relevant 'n' values must be clearly stated in the main text, figures and tables.
- The most appropriate summary statistic (e.g. mean or median and standard deviation) must be used. Standard Error of the Mean (SEM) alone is not permitted.
- Exact p values must be stated. Authors must not use 'greater than' or 'less than'. Exact p values must be stated to three significant figures even when 'no statistical significance' is claimed.

- Please include an Abstract Figure file, as well as the Figure Legend text within the main article file. The Abstract Figure is a piece of artwork designed to give readers an immediate understanding of the research and should summarise the main conclusions. If possible, the image should be easily 'readable' from left to right or top to bottom. It should show the physiological relevance of the manuscript so readers can assess the importance and content of its findings. Abstract Figures should not merely recapitulate other figures in the manuscript. Please try to keep the diagram as simple as possible and without superfluous information that may distract from the main conclusion(s). Abstract Figures must be provided by authors no later than the revised manuscript stage and should be uploaded as a separate file during online submission labelled as File Type 'Abstract Figure'. Please also ensure that you include the figure legend in the main article file. All Abstract Figures should be created using BioRender. Authors should use The Journal's premium BioRender account to export high-resolution images. Details on how to use and access the premium account are included as part of this email.

- Please include a full title page as part of your main article (Word) file, which should contain the following: title, authors, affiliations, corresponding author name and contact details, keywords, and running title.

Reviewing Editor's Comments:

Ethics Concerns:

Please include the ethics approval number, that participants gave their informed written consent and a statement indicating that the procedures satisfied the Declaration of Helsinki.

Thanks for your submitting your manuscript to The Journal of Physiology. I have now received comments from two independent assessors, both experts in the field. While both see merit in your work, commenting that it provides novel data, they both raise concerns that you will need to address. In particular, there are some methodological issues that you will need to attend to. In comments to me, one of the reviewers states that although the work is very interesting and seems to have been well-conducted, the paper is somewhat difficult to understand, due to a lack of detail in the methods and results. The other states that the paper reports novel data (grip responses to changes in friction during the hold phase of grasping/lifting), that are not surprising and that there are interesting details of the responses (asymmetry to decreased versus increased friction; affects of combining load and friction responses) that those working on grasp and manipulation, and robotic manipulators, will be interested. However, I encourage you to emphasise the novelty here: if it the first time that ultrasonic modulation of friction has been applied to a lifting task then these needs to be clarified and the significance of this approach detailed.

Referee #1:

Willemet et al present an interesting study on grip force control with changes in friction. It is novel and well-written. It is a short paper, but at times lacks important details, motivations, and justifications throughout the methods and results. Overall, it would be difficult for another person to replicate the study and analyses given the current content. Further, although it is somewhat in the realm of the Journal of Physiology, the work is rather distant from physiological mechanisms and this point could be brought out more in the paper.

In all, I think the authors over-generalize their work, as they only test a limited number of factors in a domain that could have many, many factors. To generalize to such a large extent, you would need to consider factors such as different levels of changes in friction, ease of grasp of an object, and the shape/size/texture/compliance/weight of object, never mind individual differences. All of these may give different results. Overall, it seems like the objective of your participants is to avoid slip in these experiments. You could frame your work better with this in mind and how it applies to everyday life.

I am not quite sure why the participants watched a nature documentary during the task. You justify this better in the discussion, so I think you could move some of the justification to the methods, to really insist that you did not want participants to concentrate on the task. However, in the discussion, I think you should also add that in everyday life, people will have varying levels of attention to lifting objects, therefore this should be taken into account in your interpretation and generalization of your work.

The methods lack a real section on data analysis and statistics. Also, for your t-tests, did you correct for multiple comparisons?

The results need more explanation, as it is not clear how you have analyzed your data or exactly how/why you have split the participants into groups. For the groups, you identified one 'responder' group, then split this again. This is not well-explained and makes it difficult to really conclude on what the results are or why the participants were post-experiment selected like this.

The figures are very small and could be made clearer, in terms of size and labeling.

There are a number of references and discussions in the results, which distract from the main messages. It may be best to remove some of these and save them for the discussion. Overall, it is rather difficult to actually understand the results with regard to the aims of the work.

Minor point: at the end of the introduction, you say 'For the first time...'. You do not know if you are really the first to do this and it does not add novelty to your work. You could remove it.

Referee #2:

This paper reports experiments primarily aimed at determining if humans respond with adjustments of grip force to changes in the friction at the grasp points with the object when the friction is altered after the object is grasped. These are novel data using newly developed means for accomplishing this frictional manipulation (vibration of the gripped surfaces). The results provide additional information on the behavior of humans when encountering such frictional manipulations with and without concurrent perturbations of vertical load on object. The authors speculate on sensory and integrative mechanisms for detecting and responding to these perturbations. In addition to basic research, the results may have implications for the design of virtual/augmented reality simulators and for robotic grip and manipulation in general (although these implications were not discussed).

Comments and Recommendations

Most Important

1. Statistics: While the inferential statistical comparisons all used parametric tests, some of the results (latencies) were reported using medians and quartiles. This begs the question of whether data were tested for normality, and if the use of parametric statistics is warranted. I can accept the lack of power analysis and effect sizes due to the analyses primarily based on within-subject differences from ensemble averaged data across conditions, which is a standard analysis in this

type of research over perhaps 60 years.

2. Latencies established from analysis of the grip and load signals are sometimes using the grip force signal (Figure 3) and then later (page 12) using the second derivative of grip force. Measuring latencies from kinematic and kinetic data typically use the first or second derivative (for sensitivity). Why was grip force and not grip force rate used to establish latency of response to friction changes at lift onset? This seems important because the median of 331 ms is very long (by about 3X) compared to previous reports of grip force rate responses when a subject who has been lifting an object with high friction unexpectedly encounters an object with a lower friction at the grip points.

3. It is clear how decreases in friction were achieved (vibration of the glass plates), but how was friction increased during the hold phase? Unless I missed it, this wasn't described.

4. Figure 1: The time between contacting the object and the beginning of the lift seems unusually long (the 'pre-load' phase to use Johansson and Westling's terminology). This figure seems to show a pre-load phase 3X longer than for a subject comfortably lifting a familiar object. Longer times can happen with subjects who behave as if they are cautious or unsure of the object. Please comment.

5. Page 9: Claimed absence of reflex responses to vertical loading of the object. The authors conclude that because the object's vertical position (after the load perturbation) wasn't returned to its preload position that 'reflexes were not triggered.' This speculation is probably wrong based on decades of research showing that when these 'reflexes' are activated, they seldom return the affected limb to the original position. Sherrington's idea of 'load compensation' by stretch reflexes is never achieved perfectly. Without EMG, one way to demonstrate this would be to analyze position and/or load data (higher derivatives of each with respect to time) for alterations in trajectory consistent with various reflex loops.

6. Friction perturbations during hold phase. On page 9 "The increase in friction did not induce an immediate grip force response...." This use of the word "immediate" is confusing. Do the authors mean 'at moment that the friction increase?' which would seem impossible. Or do they mean 'at some latency after the onset of the perturbation?' Please clarify or delete the word "immediate." The following sentence "Thus, frictional change was potentially not perceived by the motor control system" doesn't follow from the absence of grip force adjustments because the alternative is that the frictional change was encoded at the skin, but not acted upon. The better discussion occurs later, when the authors note how changes in skin stress/strain might not occur for increases of friction during the hold phase, versus decreases of friction.

7. The long grip force response latencies to increased load (compared to previous work) raise the question of the load force rates that were achieved by the perturbation. Were the servo motors sent a step-shaped signal and if so, how close to a step response in load was achieved? Previous work has shown that grip force rate response latency to a load perturbation is directly related to the rate at which the load changed. The shape of the load perturbation must be reported in Methods.

Less Important.

8. Throughout the manuscript the authors use terms like 'perceive' when describing motor system responses to perturbations of friction and load (E.g., Abstract). The latencies of responses to object loading and friction changes reported by Johansson and colleagues, and many others, are faster than reaction time responses and imply mechanisms other than perception (sensory encoding processed by levels of the nervous system outside of those for conscious perception).

9. Throughout manuscript: The authors should sharpen their use of terms like 'incipient slips' versus when they are actually describing stress/strain changes in the skin contacting the object. For example, top of page 3 (first full paragraph) versus 3rd full paragraph where they describe responses before 'lateral movement' and thus unrelated to 'incipient slips.'

END OF COMMENTS

Grip force control under sudden rapid change of friction

Laurence Willemet, Felix Roël, David Abbink, Ingvars Birznieks, Michaël Wiertelowski

Reviewers' remarks are shown in light gray, while our responses are printed just below them. The changes are detailed in italic in this file and highlighted in blue in the manuscript.

Reviewing Editor's Comments:

Ethics Concerns:

Please include the ethics approval number, that participants gave their informed written consent and a statement indicating that the procedures satisfied the Declaration of Helsinki.

Revision: *The study was approved by the ethics committee of the TU Delft (HREC 1722) and the study conformed to the standards set by the Declaration of Helsinki, except for the registration in the database. Participants gave their informed written consent before participating in the experiment.*

Thanks for submitting your manuscript to The Journal of Physiology. I have now received comments from two independent assessors, both experts in the field. While both see merit in your work, commenting that it provides novel data, they both raise concerns that you will need to address. In particular, there are some methodological issues that you will need to attend to. In comments to me, one of the reviewers states that although the work is very interesting and seems to have been well-conducted, the paper is somewhat difficult to understand, due to a lack of detail in the methods and results. The other states that the paper reports novel data (grip responses to changes in friction during the hold phase of grasping/lifting), that are not surprising and that there are interesting details of the responses (asymmetry to decreased versus increased friction; effects of combining load and friction responses) that those working on grasp and manipulation, and robotic manipulators, will be interested. However, I encourage you to emphasise the novelty here: if it is the first time that ultrasonic modulation of friction has been applied to a lifting task then these need to be clarified and the significance of this approach detailed.

We would like to thank the editor and the reviewers for their valuable feedback and comments. We have taken the comments into account, and we provide in this document our responses and the description of the changes that have been made in the manuscript.

Most importantly, we added details in the Materials and Methods, including a section about Data processing and statistical analyses. We also revised the statistical tests used to assess the conclusions presented in this work. For the influence of friction during lifting, we corrected for multiple comparisons and we changed the tests for non parametric ones when the variable was not normally distributed. Finally, we clarified the novelty of the work especially from neuroscience and motor control perspective in both the Introduction and Discussion sections.

Reviewer #1:

Willemet et al present an interesting study on grip force control with changes in friction. It is novel and well-written. It is a short paper, but at times lacks important details, motivations, and justifications throughout the methods and results. Overall, it would be difficult for another person to replicate the study and analyses given the current content. Further, although it is somewhat in the realm of the Journal of Physiology, the work is rather distant from physiological mechanisms and this point could be brought out more in the paper.

We are delighted to read your positive assessment and thank you for your comments. We added details in the method section, in particular regarding the perturbation stimuli, data processing, and statistical analyses such that the work is replicable by others.

We believe that the results of the paper will be of interest to the readers of the Journal of Physiology as it contributes to fundamental understanding of sensorimotor control and dexterity of the human hand. Our study reveals the mechanism behind the fine control of fingertip force, fundamental when handling objects or when using tools. Such knowledge of physiology of sensory and motor systems controlling the hand during everyday dexterous tasks is also crucial to identify functional deficits and attribute them to specific pathologies and thus enable selection of appropriate rehabilitation approaches.

We have made revisions throughout the Introduction and Discussion section to better emphasize the relevance to sensori-motor control.

In all, I think the authors over-generalize their work, as they only test a limited number of factors in a domain that could have many, many factors. To generalize to such a large extent, you would need to consider factors such as different levels of changes in friction, ease of grasp of an object, and the shape/size/texture/compliance/weight of object, never mind individual differences. All of these may give different results.

While it is true that the magnitude of grip forces are influenced by texture, object's size or shape, the main factors that contribute to the regulation of grip force are the friction of the surface and the load forces acting upon fingertips. Moreover they can change during manipulation and need sensory monitoring. Our study focuses on how frictional change when holding an object influences grip force control. We do not generalize our findings to changes other than the object's friction and load. We agree that other factors might influence the requirement for an adequate level of grip force. However regardless of these other factors, the principles of the grip force regulation remain the same and must take into account the changes in friction and load to avoid that the object slips out of hands. In the manuscript we also comment on individual differences between subjects.

What sets this study apart from others is that our apparatus allows us to change friction not just between trials, but also during the trial while the object is in hand, which was previously technically impossible. The reaction to friction change is physiologically highly relevant because friction is the fundamental force that holds objects stable. Rapid change of friction constitutes a considerable challenge for the tactile sensory system as certain mechanical events, which can readily signal frictional properties of the surface at the initial contact with the object, are not available after the object has been already gripped and held in air. Therefore we believe that our experiments have brought significant advances in our

knowledge of sensorimotor control of fingertip forces ensuring grip safety during dexterous manipulation.

Moreover, most of the previous studies on the topic were not technically able to accomplish a change of friction without changing the material of the surface (which results in a change of textures). Thus, the novelty of our study resides also in the fact that we can change the friction coefficient of the surfaces independently of the other tactile features of the surface which might provide an expected friction change cue.

Revision:

Introduction section:

“The friction between fingertip skin and the object during manipulation can vary due to several factors such as sweating and condensation. Our nervous system must adapt to this highly variable frictional interaction and react to sudden changes in friction. Despite dynamic changes in various parameters to be controlled during dexterous manipulation, humans display a remarkable ability to precisely adjust the force exerted on the object, taking into account the weight and surface properties of the object, as well as the nature of the task. The control of grasp stability encompasses the prevention of accidental slips and the prevention of excessive forces.”

“During holding an object, the nervous system is constantly regulating grip force to avoid slippage. It is proposed that it does so by detecting partial slips and monitoring the fraction between the contact area in safe contact with the skin and the area subjected to slippage, denoted as the ‘stick ratio’ (Tada, 2006; Khamis et al., 2014b; Delhayé et al., 2024).”

“Another advantage of changing friction while the participant is holding an object is that it allows us to observe adjustments to sudden relative friction change more clearly than if comparison would have been done between trials subjected to unpredictable frictional variations. Frictional adjustments within one trial are physiologically relevant as friction can significantly change between the moment when initial touch occurs (Tomlinson et al., 2009) and later stages of manipulative task (André et al., 2010). From the sensory systems point of view, the adjustments to frictional change while holding an object are more challenging without the contribution of reaching and grasping movement kinematics to friction sensing (Afzal et al., 2024).”

Overall, it seems like the objective of your participants is to avoid slip in these experiments. You could frame your work better with this in mind and how it applies to everyday life.

Thank you for the suggestion. We now have expanded the narrative linking everyday life tasks with mechanisms investigated in this study. The control of grasp stability entails both the prevention of accidental slips resulting in loss of object and the prevention of excessive fingertip forces (Johansson and Flanagan, 1989). Unnecessary high forces can make everyday life tasks hard to perform and may damage brittle objects such as fine glass. They would hinder the handling of many items such as soft or brittle food items. In the manuscript we discuss how slippage is detected and used for motor control. We introduce the metric called stick ratio, which is believed to be used by the nervous system when avoiding slip.

We would like to emphasize that our study demonstrates that the full slippage of the object is not required to trigger the responses. We observe increases of the grip forces regardless of whether we observe a gross displacement of the object or not. We have now further clarified the discussion of the sensory mechanisms and motor control strategies involved throughout the manuscript.

Revisions:

We added a description in the Introduction section:

“Peeling a juicy peach without squashing it and preventing it from slipping and falling out of hands can be a challenge especially when juice is making the surface more slippery. The friction between fingertip skin and the object during manipulation can vary due to several factors such as sweating and condensation. Our nervous system must adapt to this highly variable frictional interaction and react to sudden changes in friction. Despite dynamic changes in various parameters to be controlled during dexterous manipulation, humans display a remarkable ability to precisely adjust the force exerted on the object, taking into account the weight and surface properties of the object, as well as the nature of the task. The control of grasp stability encompasses the prevention of accidental slips and the prevention of excessive forces. A key aspect of this control involves adjusting grip forces to the friction such that a destabilizing load force, pulling the object out of the grip, is counteracted by the frictional force.

[...]

During holding an object, the nervous system is constantly regulating grip force to avoid slippage. It is proposed that it does so by detecting partial slips and monitoring the fraction between the contact area in safe contact with the skin and the area subjected to slippage, denoted as the 'stick ratio' (Tada, 2006; Khamis et al., 2014b; Delhaye et al., 2024).”

We added text to the Discussion section:

“ ... there are no known biomechanical events at the contact that would suggest how an increase in friction could be signaled by the skin receptors under static conditions. According to a biomechanical model of friction sensing mechanisms at the fingertip (Willemet et al., 2021), the friction is signaled by a skin divergence pattern. Indeed, higher friction will preclude skin from being pushed and radially diverge when pressure differences between central and peripheral portions of the contact area build up as fingertip skin is being compressed during the contact. If friction decreases, the elastic energy is released, and the skin radially diverges – an event which can be signaled by tactile afferents. However, if the skin has already diverged under a low friction condition, now increasing friction would have no effect - it would simply lock the shear strain pattern in the skin. This hypothesis is in agreement with previous experiment reporting that humans are failing to perceive friction increases (Monnoyer et al., 2023). From grip force adjustment standpoint, it is still possible that the motor system could employ a strategy that after a safe grip is established the grip force strength might be slowly released (Johansson and Westling, 1987) until afferents would start signaling small slips at the periphery of the contact area (Schiltz et al., 2021; Khamis et al., 2014b). This would make sense biologically, because it has been demonstrated that sweating might purposefully increase friction over time and thus, would be logical to expect a mechanism taking advantage of it. Nevertheless, participants in this study did not seem to exploit such a strategy and grip force was kept stable over the time course of 8 seconds after the friction change.”

“We would like to emphasize that the full slippage of the object was not required to trigger the responses because we observed that grip forces increased regardless of whether we observed a gross displacement of the object or not. Upon application of a load perturbation, the vertical position of the object changed. However, regardless that the grip force response was triggered, no response which would aim to return the object to its original position was observed. A large part of the object movement might have taken place because of stretching the skin and inducing partial slips within the contact area (Khamis et al., 2014b).”

“It is likely that frictional changes, when already holding an object, primarily activates tactile receptors sensing shear strain changes within the contact area (Macefield et al., 1996; Khamis et al., 2014a). However, the load force perturbation would stretch the skin and activate receptors also outside the contact area between its border and the nail (Birznieks et al., 2009). The load force increase and the friction reduction activate afferents in a different pattern. The load increase creates a gross deformation of the fingertip skin stretching it (sensed by SA-II afferents) and also alters the magnitude of the local strain pattern at the contact area, sensed by SA-I and FA-I afferents. In contrast, when friction is changed, this gross deformation does not occur and therefore the activation of SA-II would be lower. In addition, load force changes might activate proprioceptors (Macefield and Johansson, 1996). It has been demonstrated that when cutaneous input is not available, proprioceptors might contribute to grip reflex, but at longer latencies (Häger-Ross and Johansson, 1996). In our study, we did not observe any evidence pointing to significant contribution of proprioceptive afferents - with friction change, there was no object position change capable of activating proprioceptors. When load force perturbation was applied, no signs of myotatic reflex counteracting object’s movement were observed and response latencies were even shorter than for responses triggered exclusively by tactile afferents in friction change condition.”

I am not quite sure why the participants watched a nature documentary during the task. You justify this better in the discussion, so I think you could move some of the justification to the methods, to really insist that you did not want participants to concentrate on the task. However, in the discussion, I think you should also add that in everyday life, people will have varying levels of attention to lifting objects, therefore this should be taken into account in your interpretation and generalization of your work.

Attention is indeed an important factor when studying grip reflex. Previous studies measuring the grip force reaction strongly suggest that the sensorimotor adjustments of grip force to friction and load perturbations could be mediated subcortically (Matthews, 1984). In our experiments, we wanted to focus on reflex loops and we found that watching a documentary was diverting participants’ attention from the manipulative task and thus minimized cognitive influences on motor performance. We revised the text according to the reviewer’s suggestions.

Revision:

In the Methods section:

“During pilot experiments, we noticed that participants could not help but pay too much attention to the complex looking instrumented object. To be able to observe and study reflexive behavioral reactions we diverted their attention asking participants to watch a nature documentary while manipulating the object. The participants wore headphones

playing the audio of the documentary which the participants watched on the screen to divert their attention from the task and later were quizzed on.”

In the Discussion section:

“In this study, we wanted to reduce the influence of cognitive processes to study the response in a context close to daily life manipulation, where attention is not directed to the fine-tuning of grasping forces. After finishing their experiment, only a few participants reported perceiving the friction perturbation and, interestingly, they described the sensation as an increase in moisture. The fact that most participants were not aware of friction changes, confirmed that adjustments observed in our study do not require cognitive attention. We also conclude that watching the documentary in our study successfully helped to divert attention of participants.”

The methods lack a real section on data analysis and statistics. Also, for your t-tests, did you correct for multiple comparisons?

We added a section about Data processing and statistical analyses in the methods section. Regarding the correction for multiple comparison, we first performed a two-sample sequential t-test [1]. We found that a total of 12 participants applied a significantly different grip force when the friction was high and when the friction was low (two-sample sequential t-test). We computed the first instant of significance between these conditions by doing successive t-test. The graph below shows the minimum grip force at significance for the 12 participants who applied a significantly different grip force when the friction was high and when the friction was low (two-sample sequential t-test):

Eleven of them present adjustments before the lift off and one of them adjusts its grip force after the lift off. The latter was considered an outlier. We amended the Figure 3B with the minimal grip force at this first instant of significance for the 11 participants that accepted the H1 hypothesis before the object lift-off.

[1] Hajnal, J. (1961). A two-sample sequential t-test. *Biometrika*, 48(1/2), 65-75.

Revision:

In the Methods section we clarify:

“Initial grip force adjustments to friction before the lift-off.

To evaluate how early during the trial the grip force began to significantly differ between two frictional conditions, a two-sample sequential t-test (Hajnal, 1961) was performed comparing the grip force applied by participants during the lifting for both frictional conditions (neutral and reduced). When the t-test decision accepted the H1 hypothesis, we performed successive t-tests on grip force level measured at each 1.6 ms time bin before the object lift-off. The grip force was considered significantly different between both friction conditions if the p-value was lower than 0.05.

Grip force adjustments after perturbation.

To quantify the grip response to combined changes in friction and load force, we computed the average relative grip force changes as follows:

$$\Delta F_n(t) = \frac{F_n(t)}{1/t_p \int F_n(\tau) d\tau} - 1 \quad (1)$$

where t_p is the time at which the perturbation is applied. For each trial, we computed the average magnitude of the relative grip force adjustment as the mean of the grip force that is applied between 100 ms after the perturbation and the end of the holding phase which was 10 s long. This 100 ms latency gives a provision for potential grip reflexes to be triggered (Johansson and Westling, 1987; Cole and Abbs, 1988). The relative grip force response was then normalized by dividing it by the baseline grip force before the perturbation. To investigate the influence of the perturbation on the motor command, we performed a statistical analysis (t-test) in which the normalized grip force after the perturbation was tested in comparison to the normalized grip force during the reference condition, where no perturbation was applied.

Data distribution normality was tested with the Anderson-Darling test. If the null hypothesis was rejected, non-parametric tests were used: the t-test was replaced with a Wilcoxon signed-rank test and the ANOVA was replaced by a Kruskal Wallis H test.”

The results need more explanation, as it is not clear how you have analyzed your data or exactly how/why you have split the participants into groups. For the groups, you identified one 'responder' group, then split this again. This is not well-explained and makes it difficult to really conclude on what the results are or why the participants were post-experiment selected like this.

We removed the separation into groups in the results section since only one subject of the responder group (N=12) adjusted its grip force 0.5s after the lift off, on the contrary the 11 other subjects adjusted their grip force before the lift off. Thus, we consider this subject to be an outlier of our study.

Revisions:

In the Results section:

“We found that the surface friction significantly influenced the grip force exerted by 12 subjects out of 20 ($p < 0.05$ two-sample sequential t-test). When looking at how they lifted the object under low and high friction conditions, we found that one of them adjusted its grip force after the lift-off and was considered as an outlier; the remaining eleven participants are

shown in Fig. 3B. The grip force adjustments to friction could be observed before the lift-off after a median time of 363 ms (94-913 ms quartiles) measured from the initiation of contact. The grip force measured at the time point where high and low friction has significantly different values was 0.18 N (0.10-0.58 N quartiles). The remaining eight subjects employed significantly higher grip forces at the time of the object lift-off ($t(89) = 2.2, p = 0.03$). At the object lift-off, the grip forces measured were 3.4 ± 0.85 N, compared to 2.6 ± 0.57 N for the subjects from the responder group. These grip forces would be sufficient to maintain a safe grip regardless of how slippery the object was.”

In the Discussion section:

“The importance of the initial contact in friction sensing is supported by an observation that in 11/20 participants who adjusted their grip force to friction before object lift-off, adjustments to friction were detected at very small grip force levels being just 0.18 N on average.”

The figures are very small and could be made clearer, in terms of size and labeling.

Thank you for your comment, we increased the size of the figures as well as the labeling size.

There are a number of references and discussions in the results, which distract from the main messages. It may be best to remove some of these and save them for the discussion. Overall, it is rather difficult to actually understand the results with regard to the aims of the work.

Revision: We removed the references from the results section.

Minor point: at the end of the introduction, you say 'For the first time...'. You do not know if you are really the first to do this and it does not add novelty to your work. You could remove it.

Ultrasonic modulation of friction is a unique technology and only few research groups globally, whose work we know quite well, have access to this technology. There is no published work we could identify which would use this technology to investigate grip force adaptation mechanisms to frictional change while holding an object in the air, for the simple reason that including this technology in a handheld object is a technological feat that we are pioneering. Nevertheless, we removed this statement as suggested.

Revision: We removed 'For the first time'.

Reviewer #2:

This paper reports experiments primarily aimed at determining if humans respond with adjustments of grip force to changes in the friction at the grasp points with the object when the friction is altered after the object is grasped. These are novel data using newly developed means for accomplishing this frictional manipulation (vibration of the gripped surfaces). The results provide additional information on the behavior of humans when encountering such frictional manipulations with and without concurrent perturbations of

vertical load on object. The authors speculate on sensory and integrative mechanisms for detecting and responding to these perturbations. In addition to basic research, the results may have implications for the design of virtual/augmented reality simulators and for robotic grip and manipulation in general (although these implications were not discussed).

Thank you for this observant summary.

Comments and Recommendations

Most Important

1. Statistics: While the inferential statistical comparisons all used parametric tests, some of the results (latencies) were reported using medians and quartiles. This begs the question of whether data were tested for normality, and if the use of parametric statistics is warranted. I can accept the lack of power analysis and effect sizes due to the analyses primarily based on within-subject differences from ensemble averaged data across conditions, which is a standard analysis in this type of research over perhaps 60 years.

Thank you for the comment. In the revised manuscript we now report normal distribution testing to justify use of parametric or non-parametric tests accordingly.

Revisions:

In the Methods section:

“Data distribution normality was tested with the Anderson-Darling test. If the null hypothesis was rejected, non-parametric tests were used: the t-test was replaced with a Wilcoxon signed-rank test and the ANOVA was replaced by a Kruskal Wallis H test.”

In the Results section:

no statistical differences in object's vertical position after the friction increased or decreased (Wilcoxon signed-rank, $W = 927$, $z = 0.088$, $p = 0.93$ and $W = 883$, $z = -0.24$, $p = 0.81$)

[...]

its vertical position did not change when friction and load force were both decreased (Wilcoxon signed-rank $W = 1033$, $z = 0.87$, $p = 0.39$)

[...]

the load force change has a significant influence on both the maximum grip force rate and the reaction time (Kruskal Wallis H test, $H(2) = 25.3$, $p = 3.2e^{-6}$ and $H(2) = 11.7$, $p = 0.003$, respectively)

2. Latencies established from analysis of the grip and load signals are sometimes using the grip force signal (Figure 3) and then later (page 12) using the second derivative of grip force. Measuring latencies from kinematic and kinetic data typically use the first or second derivative (for sensitivity). Why was grip force and not grip force rate used to establish latency of response to friction changes at lift onset? This seems important because the median of 331 ms is very long (by about 3X) compared to previous reports of grip force rate responses when a subject who has been lifting an object with high friction unexpectedly encounters an object with a lower friction at the grip points.

The decision whether to use grip force or derivative was determined by the intent behind these measures. The first intent was to find the time instant where the grip forces under the

two frictional conditions became significantly different from each other, reflecting the time it takes for the grip force adjustments to take an effect after initial contact. We agree that the derivative could be also used, but in the case of our data, we found that the grip force magnitude signal was more suitable and analyses provided more reliable outcomes. After the object was lifted, grip force responses to perturbation were sudden and therefore better suited for analyses using second derivatives.

The longer latencies in comparison to some other studies can be explained by various factors including overall slower grip force increase and longer load force increase phase before the object lifted off (see also response to subsequent question 4 below).

3. It is clear how decreases in friction were achieved (vibration of the glass plates), but how was friction increased during the hold phase? Unless I missed it, this wasn't described.

Thank you for noticing this omission.

Revision:

We added the following paragraph to the Methods section:

“Before the lifting phase, the vibrations amplitude was set to either 0 or 4.5 μm . Then, the friction was either increased, decreased or kept constant by decreasing the vibrations amplitude to 0 μm , increasing to 4.5 μm , or keeping it constant, respectively.”

4. Figure 1: The time between contacting the object and the beginning of the lift seems unusually long (the 'pre-load' phase to use Johansson and Westling's terminology). This figure seems to show a pre-load phase 3X longer than for a subject comfortably lifting familiar objects. Longer times can happen with subjects who behave as if they are cautious or unsure of the object. Please comment.

This is an insightful observation. Indeed, a person will be more cautious performing grip and lift movement when handling fragile objects or when high precision/safety is required. Our instrumented object packed with high technology components would cause more cautious behavior. This is one of the reasons why we tried to divert participant's attention from the task and minimize this effect by asking them to watch a nature documentary.

In our study we did not separate the **preload and loading phase**. Note that the pre-load phase is when load is not present yet; and the loading phase is when load and grip force increase together before the object lifts off. We found that the average delay between first contacting the object and the lift off was 388 ± 248 ms. In Johansson & Westling study (1984), the time between initial contact and lift off is not explicitly reported but was in the range of 300-500 ms judging by Fig 2A and Fig 3A. Frossberg et al. (1990) reported a loading phase of 281 ± 119 ms for healthy adults.

A recent study reported slightly shorter load phase time (212 ± 7 ms) than previously reported (Delhaye et al 2024). Multiple factors such as the object's geometry, weight, inertia, texture and possibly instructions given to participants might have influenced the way subjects handle the object. We are now discussing these variations in the Discussion section.

Revisions:

In the Methods section:

“During pilot experiments, we noticed that participants could not help but pay too much attention to the complex looking instrumented object. To be able to observe and study reflexive behavioral reactions we diverted their attention asking participants to watch a nature documentary while manipulating the object.”

In the Discussion section:

“... the grip force adjustments to friction before the lift-off, were not present in all subjects and the latency was longer than previously reported in studies originating from Johansson’s laboratory (Johansson and Westling, 1984) using textured surfaces and on par with other similar recent studies using smooth glass surfaces (Delhaye et al., 2024). Apart from texture, the difference might be also caused by a lower coefficient of friction producing a less pronounced skin stretch, responsible for triggering the response. The time between the first contact and the lift-off (388 ± 248 ms) was also slightly longer, possibly being due to increased care and cautiousness participants showed when manipulating the unusual object equipped with many highly technological components.”

We also reviewed the example trial plotted in Figure 1 and changed it to one which better represents the quantitative analyses data.

5. Page 9: Claimed absence of reflex responses to vertical loading of the object. The authors conclude that because the object’s vertical position (after the load perturbation) wasn’t returned to its preload position that ‘reflexes were not triggered.’ This speculation is probably wrong based on decades of research showing that when these ‘reflexes’ are activated, they seldom return the affected limb to the original position. Sherrington’s idea of ‘load compensation’ by stretch reflexes is never achieved perfectly. Without EMG, one way to demonstrate this would be to analyze position and/or load data (higher derivatives of each with respect to time) for alterations in trajectory consistent with various reflex loops.

Thank you for the comment. In our study, in fact, we didn’t observe any compensatory reaction attempting to correct for position change. The position of the object remained unchanged after perturbation indicating that no reflex response followed. We now use more precise wording to explain this in the text.

Revisions:

In the Results section:

“After the object vertical position changed due to the load force perturbation, no corrective response aimed to minimise deviation was observed. Hence, the vertical displacement is most likely due to the object movement within the grip rather than the movement of the hand or a whole arm at the extent which would evoke corrective responses such as myotatic reflex driven by proprioceptive input.”

In the Discussion section:

“In addition, load force changes might activate proprioceptors (Macefield and Johansson, 1996). It has been demonstrated that when cutaneous input is not available, proprioceptors might contribute to grip reflex, but at longer latencies (Häger-Ross and Johansson, 1996). In our study, we did not observe any evidence pointing to significant contribution of proprioceptive afferents - with friction change, there was no object position change capable of activating proprioceptors. When load force perturbation was applied, no signs of myotatic reflex counteracting object’s movement were observed and response latencies were even shorter than for responses triggered exclusively by tactile afferents in friction change condition.”

6. Friction perturbations during hold phase. On page 9 "The increase in friction did not induce an immediate grip force response...." This use of the word "immediate" is confusing. Do the authors mean 'at moment that the friction increases?' which would seem impossible. Or do they mean 'at some latency after the onset of the perturbation?' Please clarify or delete the word "immediate."

We used the adjective 'immediate' to mean reflex response triggered by sensory input to distinguish it from possible fluctuations of grip force over longer time. For example, it might be that humans might slowly loosen the grip over time until partial slip would occur precluding further grip force decrease as proposed by Johansson and Westling (1987 page 153) and explained in the discussion of the current manuscript. In our study the latter wasn't apparent either. We now have removed ambiguous reference to 'immediate' response.

Revisions:

In the Results section:

“The increase in friction did not induce any grip force response within 8s after the perturbation.”

In the Discussion section, we explain slow grip force adjustment strategy:

“From grip force adjustment standpoint, it is still possible that the motor system could employ a strategy that after a safe grip is established the grip force strength might be slowly released (Johansson and Westling, 1987) until afferents would start signaling small slips at the periphery of the contact area (Schiltz et al., 2021; Khamis et al., 2014b). This would make sense biologically, because it has been demonstrated that sweating might purposefully increase friction over time and thus, would be logical to expect a mechanism taking advantage of it. Nevertheless, participants in this study did not seem to exploit such a

strategy and grip force was kept stable over the time course of 8 seconds after the friction change.”

The following sentence "Thus, frictional change was potentially not perceived by the motor control system" doesn't follow from the absence of grip force adjustments because the alternative is that the frictional change was encoded at the skin, but not acted upon. The better discussion occurs later, when the authors note how changes in skin stress/strain might not occur for increases of friction during the hold phase, versus decreases of friction.

We removed this statement from this location and moved to where it fits better together with the explanation given in the Discussion section.

Revision:

In the Discussion section:

“Such a result was expected because there are no known biomechanical events at the contact that would suggest how an increase in friction could be signaled by the skin receptors under static conditions. According to a biomechanical model of friction sensing mechanisms at the fingertip (Willemet et al., 2021), the friction is signaled by a skin divergence pattern. Indeed, higher friction will preclude skin from being pushed and radially diverge when pressure differences between central and peripheral portions of the contact area build up as fingertip skin is being compressed during the contact. If friction decreases, the elastic energy is released, and the skin radially diverges – an event which can be signaled by tactile afferents. However, if the skin has already diverged under a low friction condition, now increasing friction would have no effect - it would simply lock the shear strain pattern in the skin.”

7. The long grip force response latencies to increased load (compared to previous work) raise the question of the load force rates that were achieved by the perturbation. Were the servo motors sent a step-shaped signal and if so, how close to a step response in load was achieved? Previous work has shown that grip force rate response latency to a load perturbation is directly related to the rate at which the load changed. The shape of the load perturbation must be reported in Methods.

Both the load force and the friction modulation signal amplitude changes were ramped in 0.05 s. This corresponds to load force rates of 10 N/s. This rate is in the same order of magnitude as the one used in (1992, Johansson et al. Somatosensory control of precision grip during unpredictable pulling loads). The latencies are slightly higher than the one measured in that previous study. We believe that this difference might be caused by a lower friction coefficient and therefore less pronounced skin stretch triggering response, but comparing those numbers is difficult due to differences in methodology. Instead, we compared the latencies between a response to a load force change and a response to a friction decrease in our study.

Revisions:

In the Methods section:

“The pulling force produced by the DC motors was controlled to -0.5, 0 or 0.5 N with a 10 N/s load rate.”

In the Discussion section:

“The latency of grip force response to friction decrease was 274 ms (189-426 ms quartiles). In comparison for the increase of load force, participants reacted within 192 ms (72-408 ms quartiles) after the perturbation on average. While it is difficult to compare these numbers with the literature because of differences in methodology, we can note that the 100 ms difference measured between the reaction to a load force change and a friction change can signal a difference in processing of the tactile input. It is likely that frictional changes, when already holding an object, primarily activates tactile receptors sensing shear strain changes within the contact area (Macefield et al., 1996; Khamis et al., 2014a). However, the load force perturbation would stretch the skin and activate receptors also outside the contact area between its border and the nail (Birznieks et al., 2009). The load force increase and the friction reduction activate afferents in a different pattern. The load increase creates a gross deformation of the fingertip skin stretching it (sensed by SA-II afferents) and also alters the magnitude of the local strain pattern at the contact area, sensed by SA-I and FA-I afferents. In contrast, when friction is changed, this gross deformation does not occur and therefore the activation of SA-II would be lower”

Less Important.

8. Throughout the manuscript the authors use terms like 'perceive' when describing motor system responses to perturbations of friction and load (E.g., Abstract). The latencies of responses to object loading and friction changes reported by Johansson and colleagues, and many others, are faster than reaction time responses and imply mechanisms other than perception (sensory encoding processed by levels of the nervous system outside of those for conscious perception).

We do agree that 'sense' might be a better word than 'perceive' in this context, since perception might be interpreted as conscious perception and interpretation of the stimuli.

Revision: We corrected the wording through the manuscript.

9. Throughout manuscript: The authors should sharpen their use of terms like 'incipient slips' versus when they are actually describing stress/strain changes in the skin contacting the object. For example, top of page 3 (first full paragraph) versus 3rd full paragraph where they describe responses before 'lateral movement' and thus unrelated to 'incipient slips.'

Thank you for noticing. Where appropriate, we changed 'incipient slips' to 'partial slips'.

END OF COMMENTS

Dear Dr Willemet,

Re: JP-RP-2024-286486R1 "Grip force control under sudden change of friction" by Laurence Willemet, Felix Roël, David Abbink, Ingvars Birznieks, and Michaël Wiertelwski

Thank you for submitting your manuscript to The Journal of Physiology. It has been assessed by a Reviewing Editor and by 2 expert referees and we are pleased to tell you that it is acceptable for publication following satisfactory revision.

REVISION CHECKLIST:

We look forward to receiving your revised submission.

Yours sincerely,

Richard Carson
Senior Editor
The Journal of Physiology

REQUIRED ITEMS

- The Journal of Physiology funds authors of provisionally accepted papers to use the premium BioRender site to create high resolution schematic figures. Follow this link and enter your details and the manuscript number to create and download figures. Upload these as the figure files for your revised submission. If you choose not to take up this offer, we require figures to be of similar quality and resolution. If you are opting out of this service to authors, state this in the Comments section on the Detailed Information page of the submission form. The link provided should only be used for the purposes of this submission. Authors will be charged for figures created on this premium BioRender account if they are not related to this manuscript submission.

- Papers must comply with the Statistics Policy: https://jp.msubmit.net/cgi-bin/main.plex?form_type=display_requirements#statistics.

In summary:

- If $n \leq 30$, all data points must be plotted in the figure in a way that reveals their range and distribution. A bar graph with data points overlaid, a box and whisker plot or a violin plot (preferably with data points included) are acceptable formats.
 - If $n > 30$, then the entire raw dataset must be made available either as supporting information, or hosted on a not-for-profit repository, e.g. FigShare, with access details provided in the manuscript.
 - 'n' clearly defined (e.g. x cells from y slices in z animals) in the Methods. Authors should be mindful of pseudoreplication.
 - All relevant 'n' values must be clearly stated in the main text, figures and tables.
 - The most appropriate summary statistic (e.g. mean or median and standard deviation) must be used. Standard Error of the Mean (SEM) alone is not permitted.
 - Exact p values must be stated. Authors must not use 'greater than' or 'less than'. Exact p values must be stated to three significant figures even when 'no statistical significance' is claimed.
-

Reviewing Editor's comments:

Thank you for submitting you revised manuscript to The Journal of Physiology. I have now received reports from the original two reviewers, both of whom are satisfied with your amendments. However, before we can publish this manuscript please submit larger versions of Fig. 6 and Fig. 7.

Referee #1:

I thank the authors for their very clear replies and modifications to their paper. It is much improved, being easier to follow and it puts the work very well into perspective. I have only a further minor suggestion: to make the figures still a little bigger, especially figures 6 and 7.

Referee #2:

the authors have addressed my concerns in this revision either through adding needed context or explanations, or by more direct changes to content. They have expanded the Methods and provided more details in Results.

END OF COMMENTS

Grip force control under sudden rapid change of friction

Laurence Willemet, Felix Roël, David Abbink, Ingvars Birznieks, Michaël Wiertlewski

Reviewers' remarks are shown in light gray, while our responses are printed just below them.

Reviewing Editor's Comments:

Thank you for submitting you revised manuscript to The Journal of Physiology. I have now received reports from the original two reviewers, both of whom are satisfied with your amendments. However, before we can publish this manuscript please submit larger versions of Fig. 6 and Fig. 7.

We would like to thank again the editor and the referees for their time and valuable feedback to improve the quality of our manuscript. We provide now a new version with larger figures 6 and 7.

Referee #1:

I thank the authors for their very clear replies and modifications to their paper. It is much improved, being easier to follow and it puts the work very well into perspective. I have only a further minor suggestion: to make the figures still a little bigger, especially figures 6 and 7.

Thank you for your positive comments. We have now made the figures 6 and 7 bigger.

Referee #2:

the authors have addressed my concerns in this revision either through adding needed context or explanations, or by more direct changes to content. They have expanded the Methods and provided more details in Results.

We are delighted to read your positive assessment and thank you for your comments.

Dear Dr Willemet,

Re: JP-RP-2024-286486R2 "Grip force control under sudden change of friction" by Laurence Willemet, Felix Roël, David Abbink, Ingvars Birznieks, and Michaël Wiertelwski

We are pleased to tell you that your paper has been accepted for publication in The Journal of Physiology.

Yours sincerely,

Richard Carson
Senior Editor
The Journal of Physiology

If you would like to receive our 'Research Roundup', a monthly newsletter highlighting the cutting-edge research published in The Physiological Society's family of journals (The Journal of Physiology, Experimental Physiology, Physiological Reports, The Journal of Nutritional Physiology and The Journal of Precision Medicine: Health and Disease), please click this link, fill in your name and email address and select 'Research Roundup':

<https://www.physoc.org/journals-and-media/membernews>

- You can help your research get the attention it deserves! Check out Wiley's free Promotion Guide for best-practice recommendations for promoting your work at: www.wileyauthors.com/eoo/guide. You can learn more about Wiley Editing Services which offers professional video, design, and writing services to create shareable video abstracts, infographics, conference posters, lay summaries, and research news stories for your research at: www.wileyauthors.com/eoo/promotion.

The Corresponding Author will receive an email from Wiley with details on how to register or log-in to Wiley Authors Services where you will be able to place an order

Reviewing Editor's comments:

Thank you for submitting higher-resolution images of Figs 6 and 7. I am now satisfied that your manuscript is suitable for publication.

END OF COMMENTS